# The Effect of Different Induction Methods on the Structure and Physicochemical Properties of Glycosylated Soybean Isolate Gels

**DOI:** 10.3390/foods11223595

**Published:** 2022-11-11

**Authors:** Jiangying Yu, Baozhong Sun, Songshan Zhang, Xiaochang Liu, Peng Xie

**Affiliations:** Institute of Animal Sciences, Chinese Academy of Agricultural Sciences, Beijing 100193, China

**Keywords:** glycosylated proteins, gel structure, physicochemical properties

## Abstract

Soybean protein isolate (SPI), as a full-valued protein, is rich in nutrients, such as amino acids. However, the isolated structure of soybeans makes it difficult to react and thus prepare good gels. In order to further improve the properties of SPIs and to prepare plant-based gels with good performance, this experiment was conducted to prepare maltodextrin glycosylated soybean isolate (MGSI) by the glycosylation of SPI and maltodextrin (MD), and the gels were prepared by thermal induction, transglutaminase (TGase) induction, and TG-MgCl_2_ co-induction of this glycosylated protein to investigate the effects of different induction methods on the structure and properties of the gels produced by MGSIs. Moreover, the effects of different induction methods on the structure and properties of the gels produced by MGSI were investigated. SDS-PAGE protein electrophoresis, FTIR spectroscopy, and endogenous fluorescence spectroscopy revealed that all three inductions result in the covalent bond cross-linking of MGSI during the gel formation process. Compared with thermal induction, the TGase-induced MGSI secondary structure had a higher content of β-folded structures, increased fluorescence intensity of tertiary structures, and produced a red shift. The gel induced by TGase in collaboration with MgCl_2_ contains a more β-folded structure and irregular curl and increases the β-turned angle and α-helix content further, the endogenous fluorescence λmax is significantly red-shifted, and the fluorescence intensity increases, demonstrating that the tertiary structure of MGSI unfolds the most, forming multilayered gels with the tightest structures. The three gels were analyzed by rheology and SEM, showing that the TGase-MgCl_2_ synergistically induced gel had the highest energy-storage modulus G’, viscoelasticity, and water-holding capacity, as well as the densest gel structure. In conclusion, the combined treatment of enzyme and MgCl_2_ might be an effective way of improving the structure and gel properties of SPI. This study helps to promote the high-value utilization of SPI and the development of plant protein gels.

## 1. Introduction

Protein gels, as typical amphiphilic and stable delivery systems, have a three-dimensional mesh structure that can encapsulate hydrophilic or hydrophobic components and serve a variety of purposes, such as increasing the moisture content of food, increasing tenderness and elasticity, and protecting and stabilizing active substances [1]. The main common food proteins used to make protein gels in the research at present are oat, konjac, and whey proteins, as well as casein. Among them, soy protein isolate (SPI) is widely used in the preparation of food gels due to its low cost and rich nutritional content [2]. However, due to the dense spherical structure of SPI, the molecular groups are wrapped in the interior, which makes it more difficult to gel, and the rheological and stability properties of its gels are poor, limiting its application in food [3]. In order to improve the physicochemical properties of SPI and extend its application, the structure of the amino acid side-chain groups of SPI molecules is usually changed in the research to date, or the main chain of the protein molecule is specifically sheared for subsequent gelation, micro-encapsulation, and other productions. Common methods of modification include glycosylation, enzymatic cross-linking, hydrolysis, and deamidation [4]. Among them, the glycosylation modification process does not require any chemical catalyst, is green and healthy, and is therefore widely used in food protein modification processes [5]. The SPI modified by glycosylation covalently combines the ε-amino group of the protein molecule with the reducing carbonyl group of the sugar molecule, resulting in a glycosylated SPI with better emulsification, rheology, and structural stability than the SPI, which is more suitable for gel preparation [6].

Traditionally, protein gels have been prepared by thermal induction, but the use of transglutaminase (TG), salt ions, and other inducers can further promote the cross-linking of glycosylated proteins and facilitate the preparation of more stable gel structures [7,8]. The cross-linking of the γ-acyl group of glutamyl residues on the peptide chain with the ε-amino group, the primary amino group, and water on lysine improves the structural stability of the protein gels and increases the gel density and swelling properties [9]. Similarly, the addition of divalent cations, such as Mg^2+^, during gel formation can help to form salt bridges within the protein gel and promote structural stability [10]. In addition, the synergistic induction of TG and divalent metal ions during gel formation can significantly improve the structural stability, cross-linking, and bioavailability of the gels [11]. However, studies using three approaches to induce the preparation of glycosylated protein gels and comparing their structural analysis and physicochemical properties are rare. In this study, glycosylated protein gels were prepared by thermal induction, TGase induction, and the MgCl_2_-TGase co-induction of MGSI using soybean isolate glycosylated maltodextrin (MGSI) as the substrate, and the microstructural changes and physicochemical properties of the protein gels were investigated under different induction methods in order to provide some theoretical basis for the development and application of good performance composite gels.

## 2. Materials and Methods

### 2.1. Materials and Reagents

Soybean protein isolate (food grade), maltodextrin (food grade), and TG enzyme (enzyme-specific activity of 100 U/g) were obtained from the Solabao Reagent Company (Beijing China); all the other reagents were of commercially available analytical purity. Acrylamide (reagent pure), bromophenol blue (reagent pure), Tris (reagent pure), TEMED (reagent pure), and ammonium persulfate (reagent pure) were obtained from the Chengdu Kelong Chemical Reagent Factory, and the sodium dodecyl sulfate (reagent pure) (pure) was obtained from Chongqing Chuandong Chemical Industry (Group) Co., Chongqing, China.

### 2.2. Apparatus and Equipment

A DYCZ-26C-type protein bi-directional electrophoresis instrument: Beijing Liuyi Instrument Factory; iS5-type Fourier transform infrared spectrometer: Jin Gong Instrument; UV752N-type upper ultraviolet spectrophotometer: Hai Youke Instrument Co., Ltd., Shanghai, China; and T18 Digital ULTRATURRAX crusher: Hunan Zhongji Materials Import and Export Co., Changsha, China were used.

### 2.3. Test Methods

#### 2.3.1. Preparation of Maltodextrin Glycosylated Soybean Isolate (MGSI)

MGSI was prepared by the method of ZHANG et al. [12], with slight modifications. The MGSI was obtained by the dispersion of SPI and MD at a mass ratio of 1:2 in 0.2 mol/L phosphate-buffered solution at pH 7.0, adjusted to pH 9.0 with NaOH, in a water bath at 85 °C for 1.5 h. The MGSI was cooled to room temperature, placed in a dialysis bag with a MWCO of 3500, dialyzed for 48 h, and stored in a refrigerator at 4 °C.

#### 2.3.2. Preparation of Different Gels

(1)Preparation of heat-induced gels (HG gels): Take 5% MGSI solution, stir magnetically for 30 min, adjust pH to 6.5, heat in a water bath at 60 °C for 1 h, then cool in an ice-water bath to room temperature and leave for 24 h in a refrigerator at 4 °C.(2)Preparation of TG enzyme-induced gel (TG gel): Take 5% MGSI solution, stir magnetically for 30 min, adjust the pH value to 6.5, heat in a water bath at 55 °C, add 160 U/g of TG enzyme, and leave the reaction time for 30 min. The reaction time was 30 min.(3)Preparation of MgCl_2_-TG enzyme co-induction gel (MTG gel): Take 10% MGSI solution, stir magnetically for 30 min, adjust the pH value to 7.5, add 100 U/g TG enzyme, add MgCl_2_ to make the concentration 0.1 mol L^−1^, and then perform the reaction in a water bath at 40 °C. After the reaction, remove the samples and place in a water bath at 80 °C to inactivate the enzyme, cool rapidly to room temperature, and place in a refrigerator at 4 °C for 24 h.

#### 2.3.3. Polyacrylamide Gel Electrophoresis

The lyophilized powder of the gel was hydrated and diluted to 2 mg/mL of liquid, 12 ul was obtained, 3 ul of loading buffer and deionized water were added to make a 15 ul protein pretreatment sample, it was placed in a 95–99 °C water bath for 5 min, and it was then placed in a 4 °C refrigerator for temporary storage. A 12% concentration of separation gel (containing 30% acrylamide, 1.5 mol L^−1^ Tris, 10% SDS, deionized water, 2% ammonium persulphate, and TEMED) and a 5% concentration gel (containing 30% acrylamide, 0.5 mol L^−1^ Tris, 10% SDS, 2% ammonium persulphate, TEMED, and deionized water) were prepared. The electrophoresis conditions were constant voltages in stages, firstly, at 80 V. When the bromophenol blue had run to the intersection of the two gels, it was switched to 100 V. The electrophoresis was continued until the samples were 1 cm from the bottom of the plate and stopped. The plate was placed on a shaker and stained with Kemas Brilliant Blue staining solution until a band appeared; then, the staining solution was poured out and decolorized by adding a decolorizing solution until the background color was completely removed and the band was clear. Scanning was performed using a gel imaging system, with 3–4 parallel groups of each gel.

#### 2.3.4. Fourier Transform Infrared Spectroscopy (FTIR)

(1)To compare the intensity of the absorption peaks of different samples in the spectral scan range and to analyze the changes in cross-linked bonds between MGSIs under different induction methods, the method of Bing Zhao [13] was referred to and slightly modified. The SPI and gel content were adjusted to 40 mg/mL, the mixtures were freeze-dried, and the lyophilized powders of HG, TG, and MTG gels were mixed with potassium bromide (KBr) at a ratio of 1:100 (*w*/*w*), ground well in a mortar, made into 1–2 mm specimens with a hydraulic press, and held at 10 kg of pressure for 30 s. KBr was used as a blank, and the pressed transparent samples were placed into a Fourier transform infrared spectrometer. The FTIR spectra were smoothed and automatically corrected for the baseline using the Thermo Scientific OMNIC software, and three parallel sets of each sample were measured.(2)The secondary structure was determined by the method of Zhang [14] with modifications: the range of 1600–1700 cm^−1^ in the amide I band was obtained for the calculation; the PeakFit 4.12 software was used to correct the baseline, smoothing, deconvolution, and second-order derivative; the Gauss peak shape was used for fitting; and the residuals were minimized after several fits [15]. The ratio of the peak area of each secondary structure to the total peak area was determined as the content of the secondary structure, and the protein was obtained as a β-folded structure in the ranges of 1610–1640 cm^−1^ and 1670–1680 cm^−1^; α-helical structure in the range of 1650–1660 cm^−1^; β-helical structure in the range of 1660–1700 cm^−1^; β-turned structures; and the range 1640–1650 cm^−1^ was qualitative and quantitative information for secondary structures, such as irregularly curled structures [16].

#### 2.3.5. Endogenous Fluorescence Intensity

A phosphate-buffered solution of 50 mmol/L, pH 7.4, was prepared and added to the gel sample to produce a solution of 1.0 mg/mL MGSI. The endogenous fluorescence intensity of the sample was measured using a fluorescence spectrometer. The measurement conditions were set as follows: the excitation wavelength was set to 290 nm, the scanning emission wavelength was set to 300–400 nm, the voltage was set to 400 V, the scanning speed was set to 1500 nm/min, and the slit width for excitation and emission was set to 5 nm. Each sample was measured three times.

#### 2.3.6. Scanning Electron Microscopy (SEM)

Lyophilized gel sheets were attached to the sample platform using adhesive double-sided conductive adhesive, platinum was sprayed in vacuum to produce the samples, and the samples were observed and photographed in the SEM at 5 Kx and 20 Kx with a 10 kV electron beam.

#### 2.3.7. Emulsification Activity and Stability

A total of 0.1 g of gel product was weighed with a protein concentration of 0.2% and dissolved in 5 mL of phosphate buffer at pH 7.0. The sample solution was mixed with algal oil at a ratio of 3:1 and homogenized at 12,000 r/min for 1 min. The homogenate was immediately aspirated from the bottom of the centrifuge tube, added to the solution, shaken and mixed, and the absorbance was measured at A0. The 0.1% SDS solution was used as a blank control and the emulsification activity (EAI) and emulsion stability (ESI) were calculated according to the following Equation:(1)$$\begin{array}{c} \mathrm{EAI}=\frac{2\times 2.303\times A_{0}N}{\rho \times \left(1-\varphi \right)\times 10^{4}}\\ \mathrm{ESI}=\frac{A_{10}}{A_{0}}\times 100\% \end{array}$$
where φ is the volume fraction of the oil phase (φ = 0.2); ρ is the protein concentration/(g/mL); and N is the dilution multiple (100). 

#### 2.3.8. Rheological Properties

Referring to the method in the literature [17], the samples were prepared in distilled water to a concentration of 12% gel liquid, stirred, and dispersed well. The MCR-301 rheometer was used with parameters set in rotary mode at 25 °C, set to a plate spacing of 1 mm, a shear rate range of 0.01–1000 s^−1^, set to automatic mode, and a maximum duration of 20 min for individual tests.

Measurement of storage modulus (G’) and loss modulus (G’’): A rotational viscometer was used to measure the rheological behavior of the emulsions with the gap between the vertebral plates set to 0.80 mm at 25 °C. In the sweep mode, a constant strain of 0.5% was set, and the loss modulus was measured at frequencies from 0.1 to 10 Hz. The loss modulus and storage modulus of the gel were measured.

#### 2.3.9. Determination of the Water-Holding Capacity (WHC) of Gels

A quantity of gel (W0) was placed in an appropriate amount of distilled water and stirred for 3 h at a temperature of 30 °C with a magnetic stirrer set to 50 r/min. The swollen gel was removed, dried on filter paper, and weighed (Wt).
WHC/% = (Wt − W0)/W0 × 100%

#### 2.3.10. Determination of Gel Strength

The gel was cut into cylindrical shapes (20 mm diameter and 10 mm height) and the gel strength of the samples was determined using a TA-XT mass spectrometer. A P/36R probe was used with a trigger force of 0.0294 N, a drop height of 10 mm, a premeasurement speed of 5.0 mm/s, a test speed of 1. 0 mm/s, and a postmeasurement speed of 5.0 mm/s.

#### 2.3.11. Data Processing 

All the experiments were repeated three times in parallel, and the results are expressed as the mean of the three parallel values, with the error line indicating the standard deviation. The experimental data were fitted using Origin 8.5 and SPSS and processed for statistical analysis (*p* < 0.05) separately.

## 3. Results and Analysis

### 3.1. SDS-PAGE Protein Electrophoresis Analysis of HG, TG and MTG Gel Particles

The resulting gel particles were characterized by SDS-PAGE electrophoresis spectra after dialysis purification to reveal the difference in molecular weight of the gels during the reaction. When large molecular products are formed, staining bands appear at the junction of the concentrated and separating gels or trailing in the separating gels. As shown in Figure 1, the bands in MGSI are mainly around 11 and 245 kDa, with bands also appearing at 17, 75, and 63 kDa. This is probably due to incomplete grafting of SPI and MD in MGSI, resulting in bands of different molecular weights. The remaining three gels (HG, TG, and MTG) all showed bands mainly at 245 kDa, indicating that MGSI was further cross-linked after induction and thus increased in molecular weight, which led to its accumulation at the top of the isolated gel [18]. In the other bands, the colors of all three gels narrowed and lightened at the edges, demonstrating that the content of other molecular-weight proteins was reduced and that induction could promote cross-linking between MGSIs [19].

### 3.2. FTIR Analysis of the Different Gels

FTIR spectroscopy was used to analyze and characterize the molecular structure of proteins by comparing the differences in absorbance of different groups in their corresponding optical absorption regions and is at present widely used to verify protein–polysaccharide interactions and to assess structural changes in glycosylated proteins. The characteristic absorption bands of the IR scan include the amide I band (1600–1700 cm^−1^, H-O-H bending, and C=O stretching vibrations), the amide II band (1530–1550 cm^−1^, N-H bending), and the amide III band (1260–1330 cm^−1^, C-O and C-O) [20]. The FTIR spectra of MGSI and the three protein gel particles (HG, TG, and MTG gels) are presented in Figure 2. Ali et al. also demonstrated that TGase can promote deamidation, reduce protein hydrophobicity, improve protein interactions, and promote gel formation [15]. Compared with the uninduced MGSI protein, the absorption peaks of TG and MTG gels showed an increasing trend at 1540 cm^−1^ and 1600 cm^−1^, which indicated a decrease in partial hydrogen bonding, a change in bond length, and an increase in wave number of the N-H chemical bond of the protein under the influence of TGase. Fluctuations from 1600 cm^−1^ to 1700 cm^−1^ were mainly influenced by the stretching vibration of the C=O bond. The waveforms of MGSI proteins modified by the TG enzyme all shifted towards higher wave numbers, probably because the bond lengths of chemical bonds, such as C=O, were reduced. The stretching vibration of chemical bonds was inversely proportional to the square root of the bond length, and the reduction in bond length increased the frequency of the stretching vibration, thus leading to an increase in wave numbers [21]. The absorption peaks of MGSI protein gels were wider in the range of 3200 cm^−1^~3700 cm^−1^ compared with the three protein gels (HG, TG, and MTG) due to the increase in the number of hydroxyl groups due to the covalent bonding between the proteins, which caused an increase in the C-OH vibration [22].

The amide I band (1600–1700 cm^−1^) in the characteristic absorption spectrum of the protein can reflect the changes in the secondary structure of the protein. The absorption peaks of α-helix, β-fold, β-turn, and random curl producing stretching vibrations were selected in the ranges of 1650–1660 cm^−1^, 1610–1640 cm^−1^, 1660–1700 cm^−1^, and 1640–1650 cm^−1^, respectively. The changes in the secondary structure content of the protein can be observed in Table 1. Compared with SPI, the main structures in the three gel particles were still β-fold and α-helix, and the β-fold content showed an increasing trend. This indicates that on the one hand, induction has an effect on the secondary structure of the protein [23], and on the other hand, the introduction of the sugar chain causes the protein peptide chain to unfold, the internal groups to be exposed, and the spatial conformation of the protein to change. Among these, the β-folded structure plays a major role in maintaining the structure of the protein gel network [24]. This illustrates the enhanced structural stability of the protein in the gels following induction [13]. This also provides an explanatory basis for the improved performance of the three gels later in the paper. Compared with SPI, the proportion of β-folded structures, β-fold, and α-helices in the heated HG gel particles was increased, and the proportion of irregular coiling was reduced. The increased order and stability of the protein at this point favored the stability of the gel, but the overly stable structure was not conducive to the conformational changes required for the functional properties. Compared with the HG gel particles, the addition of TGase resulted in an increase in β-folded structures and irregular curls and a decrease in α-helices in the TG gel particles. The α-helices are mainly located inside the molecular chains, and the decrease in their content may be due to the catalytic effect of TGase on the protein, resulting in the disruption of molecular-bonding hydrogen bonds and a relatively disordered secondary structure when the protein molecular structure is fully unfolded [25]. The increase in the randomly curled structure may also be due to the unfolding of the MGSI spatial structure [14]. The main increase in MTG gels compared with HG gels was in the β-fold structure. β-folding is mainly present on the surface of proteins and can contribute to the structural stability and toughness of protein gels [26]. The reason for this may be that the addition of Mg^2+^ changes the polarity of the solution, resulting in the disruption of the internal hydrogen-bonding structure of the protein and the unfolding of the α -helix and β-turn angle in the secondary structure to a β-folded structure [27].

### 3.3. Analysis of Endogenous Fluorescence Spectra of Different Gels

In this study, the fluorescent material of the gels was examined in the range of 340–370 nm [28] to characterize the changes in the tertiary structure of MGSI proteins after induction. Typically, a red shift in the maximum absorption wavelength indicates the exposure of the fluorescence-emitting group to the solvent and unfolding of the protein molecule; a blue shift indicates that the aggregation of the protein has occurred [29]. As can be observed in Figure 3, the order of fluorescence intensity of the three gels in the scanned wavelength range is MTG gel > TG gel > HG gel > MGSI, and the fluorescence intensity at the maximum fluorescence intensity of the other three gels is higher than that of MGS. It indicates that induction has an effect on the tertiary structure of the protein. The fluorescence intensity of TG gel is greater than that of HG gel. This may be due to the gradual loosening of the MGSI structure under the action of TGase, which gradually exposed the internally wrapped color-emitting amino acids and increased the fluorescence intensity [30]. Additionally, the fluorescence intensity was increased further after the synergistic induction of TGase and MgCl_2_, probably due to the partial deamidation of glutamine and asparagine, the reduction in the spatial site block, and the exposure of the tryptophan chromophore [31]. In addition to this, all three gels underwent a small degree of red shift and an increase in peak intensity compared with the MGSI, suggesting that the combined treatment had an effect on the looser tertiary structure of the MGSI, resulting in the exposure of the chromogenic group of the protein to the solvent and an increase in fluorescence intensity [32].

### 3.4. Microstructure of Different Gels (SEM)

The three-dimensional network gel structure formed by the cross-linking of different gels after induction can be visualized by scanning electron microscopy. As shown in Figure 4, it can be observed from the SEM images that all three gels present cross-linked sheets, which are not consistent with the spherical shape of SPI due to the occurrence of the glycosylation reaction to some extent destroying the structure of the SPI, and following induction, the MSGI cross-linked with each other, thus forming a network structure. The HG gels were sparse and thin, with an uneven spatial structure and irregular pores, while the TG gels were more densely cross-linked but still coarser and with larger voids [33]. Han et al. [34] showed that the induction of TG enzyme led to a decrease in the number of small blocks in the protein gels and a gradual increase in the number of large protein cross-links, which is consistent with the results of this experiment. This demonstrates that the characterization of the gel network is significantly improved by the synergistic induction of Mg^2+^ and TG enzymes. The results are consistent with the results of Wang [35] et al.

### 3.5. Analysis of the Emulsification Properties of Different Gels

The gel particles of SPI and its modified proteins are usually amphiphilic (hydrophilic and lipophilic) and have been widely used as stabilizers in various emulsions [36]. The variation of EAI and ESI values can visualize the emulsification and stabilization ability of the gel particles as emulsifiers, indicating the absorption ability of protein molecules at the oil/water interface [37]. Figure 5 presents that the emulsions formed by the three gel particles (HG, TG, and MTG gels) as emulsifiers increase the emulsification activity and stability compared with those formed by stabilization with MGSI protein particles. This result may be due to the self-assembly of gel particles bridging adjacent droplets to form a more homogeneous microgel network-like structure that can act together at the water/oil interface to stabilize the emulsion [38]. Similar results were obtained by Jiang et al. [39]. Compared with HG gels, the addition of TGase may serve to further covalently cross-link MGSI molecules internally or intermolecularly, forming gel particles that are aggregated and have a thicker interfacial protein layer, making interaction with small-molecule oil droplets more difficult and therefore reducing emulsification. This is consistent with the study conducted by FLANAGAN et al. [20]. Compared with TG gels, MTG gels have improved emulsification activity and reduced emulsion stability. The metallic nature of Mg^2+^ promotes the exposure of hydrophobic groups in the internal structure of proteins, inducing the unfolding of protein surface and internal protein structures, while its own large hydration radius has a strong attraction to water molecules, providing a fully bound environment at the oil–water interface and enhancing emulsification activity [40]. However, in the presence of Mg^2+^ ions, the ionic bonding formed by proteins and metal ions produces a certain spatial site-blocking effect, which inhibits protein aggregation and promotes the further aggregation of droplet oil, thus leading to weaker emulsion stability. Therefore, of the three gel particles, HG gel particles are considered to be better emulsifiers.

### 3.6. Static Rheological Analysis of Different Gels

As shown in Figure 6, the apparent viscosities of TG and MTG gels were higher than those of HG gels, probably due to the “expansion” of the volume of the gel particles as a result of the destruction of the original structure and the formation of a new structure by external forces: the shear-thickening phenomenon. Previous studies have also shown that the apparent viscosity of casein gels and whey protein gels treated with enzymes or salts is greater than that of untreated protein gels [41,42].

### 3.7. Dynamic Rheological Analysis of Different Gels

The value of the energy-storage modulus (G’) represents the elastic and solid-like characteristics of the gel sample, and the value of the loss modulus (G’’) represents the mobile phase strength and fluid-like behavior of the gel sample [43]. A larger G’ and a smaller loss modulus G’’ is small, demonstrating that the fluid tends to exhibit solid-like properties. As the frequency increases, both the G’ and G’’ of the gel particles in the three gels gradually increase, and G’ is always greater than G’’. Therefore, it was confirmed that the gel particles reflected the characteristics of an elastic gel, and the characteristics of gels prepared in this study were similar to those of viscoelastic materials. This result is consistent with the results of Jiang [39] et al. who used SPI gel particles to stabilize emulsions. As shown in Figure 7, the G’ of all three gels showed a trend of first increasing and then leveling off due to the change in intermolecular interactions. Among them, the G’ of MTG gels were higher than TG gels and HG gels, probably due to the higher protein content in MTG and the formation of larger cross-links after protein aggregation. It is also possible that more covalent bonds were generated by the combined induction of TGase and Mg^2+^, forming a gel structure with a combination of covalent and ionic bonds. The higher G’ of TG gels than HG gels may be due to the TGase catalyzing the transfer of acyl groups in MGSI and SPI proteins, where the amino groups in the peptides are converted to NH_3_ and released, and a deamination reaction occurred. The exposure of surface sulfhydryl groups to the surface of the molecule results in the formation of more disulfide bonds between SPI and MGSI protein molecules, increased hydrophobicity, and enhanced gel strength [44]. The G’ and G’’ of the three gels increase with frequency, proving that these gels have some frequency dependence. Dickinson et al. [45] demonstrated that structurally stable gels are intrinsically stabilized by the cross-linking of mainly covalent bonds. Tang et al. [46] found that the increase in G’ at high-intensity frequencies could also be due to the formation of non-covalent physical bonds, such as ionic bonds cross-linking, and the G’’ of the induced gels also increased and eventually leveled off with increasing shear frequency.

Loss factor Tan δ (G’’/G’) is the ratio of loss modulus (G’’) and storage modulus (G’’), representing the relative distribution of “viscosity” and “elasticity” during the formation of protein gel [47]. As shown in Figure 7, within the test frequency range, all samples’ G’s are higher than G’’, and tan δ value is far less than 1.0, which indicates that HG gel, TG gel, and MTG gel exhibit elastic behavior and solid-like characteristics [48]. Generally speaking, the better the gel matrix is, the lower the final value tan δ is. Therefore, MTG gel is the best in the performance of gel matrix, which indicates that the addition of TG enzyme and Mg^2+^ improves the gel performance and forms a more compact and stable gel. With the increase in frequency, the tan δ value of MTG gel changes gently and the slope is close to zero and does not depend on the change in frequency, indicating that the gel has strong gel performance [49]. Tan δ value of HG gel is about 0.5–0.6, which indicates that the system is a gel characterized by weak elasticity [50]. The dependence of tan δ of the other two gels on frequency gradually increases, especially HG gel, indicating its gel structure is relatively fragile, and the network structure is vulnerable to damage in dynamic viscoelastic measurement.

### 3.8. Gel Strength and Water Retention Analysis

Gel strength and water retention are important properties of protein gels. Compared with HG gels, the gel strength and water-holding capacity of the TG and Mg^2+^-induced gels were increased. As shown in Figure 8, the addition of the TG enzyme increased the gel strength to 244.83 ± 2.67 g and water retention to 244.59 ± 5.5% compared with the thermally induced HG gels with gel strength 163.22 ± 3.42 g and water retention 135.03 ± 4.09%. This may be due to the enzymatic degradation of MGSI by the cross-linking action of TGase, exposing the lysine residues, forming smaller gel pores and more adequate cross-linking between proteins, resulting in higher gel-strength properties and water-holding capacity [51]. Compared with TG gels, the strength of MTG gels was enhanced to 356. 45 ± 6.01 g and the water-holding capacity was increased to 252.71 ± 7.3% after the addition of Mg^2+^. This may be due to the fact that Mg^2+^ divalent ions induce aggregation through electrostatic shielding, ionic/hydrophobic interactions, and cross-linking with negatively charged carboxyl groups of protein molecules to establish protein–cation–protein bridges [52]. Cross-linking between gels was enhanced further by salt bridges and covalent bonding, which were able to trap more water molecules and improve the water-holding capacity and gel strength. A good synergistic effect between Mg^2+^ and TGase treatment was demonstrated further [53].

## 4. Conclusions

The results show that MGSI is tightly bound within the gel by the combined action of TGase and Mg^2+^, with increased β-fold structure content and a stable protein tertiary structure compared with thermal induction and TGase action. In addition, the gel elasticity, water-holding capacity, and structural denseness of MTG gels are improved further by the synergistic effect of TGase and MgCl_2_. This identified the MTG gel as the best performing gel among the three gels, which can be applied to a variety of food-processing scenarios. These results suggest that the synergistic induction of Mg^2+^ and TGase may be a promising approach to improving the potential of protein gels for applications in the food processing industry.

## Figures and Tables

**Figure 1 foods-11-03595-f001:**
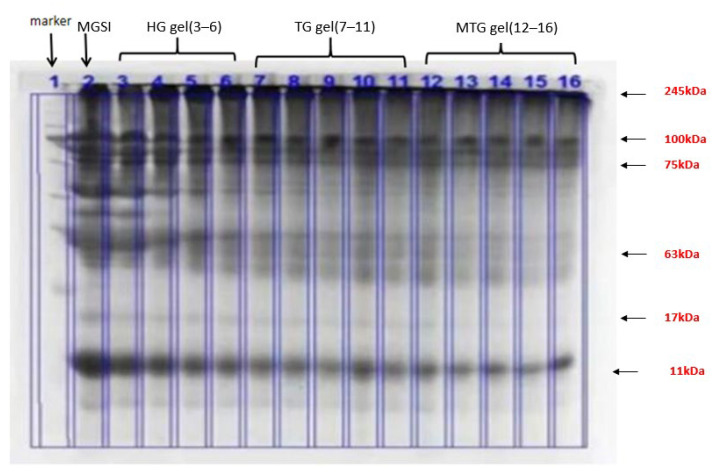
SDS-PAGE electrophoresis of the three protein gels.

**Figure 2 foods-11-03595-f002:**
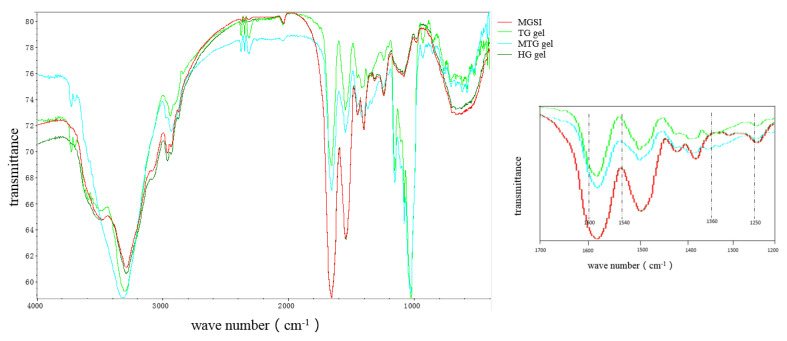
FTIR spectra of different gels.

**Figure 3 foods-11-03595-f003:**
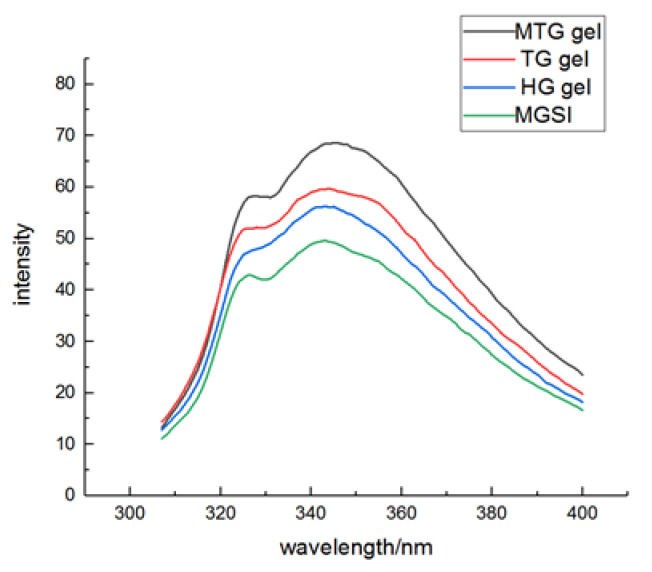
Endogenous fluorescence spectra of different gels.

**Figure 4 foods-11-03595-f004:**
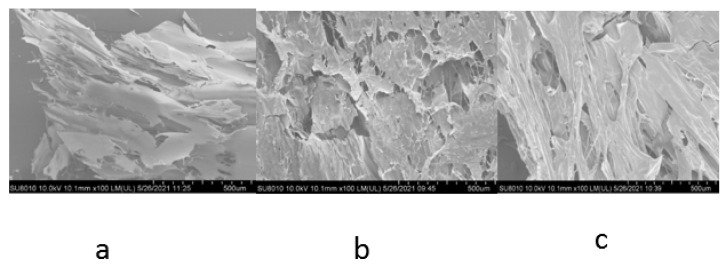
Microstructure of different gels. Note: (**a**) for HG gels; (**b**) for TG gels; (**c**) for MTG gels.

**Figure 5 foods-11-03595-f005:**
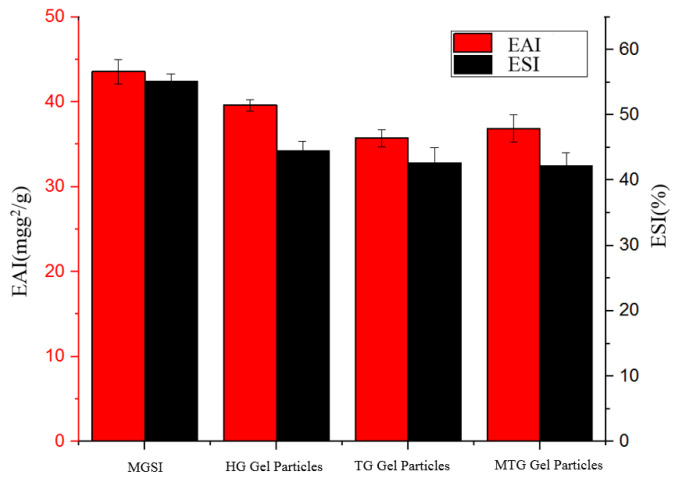
Emulsification analysis of different gels.

**Figure 6 foods-11-03595-f006:**
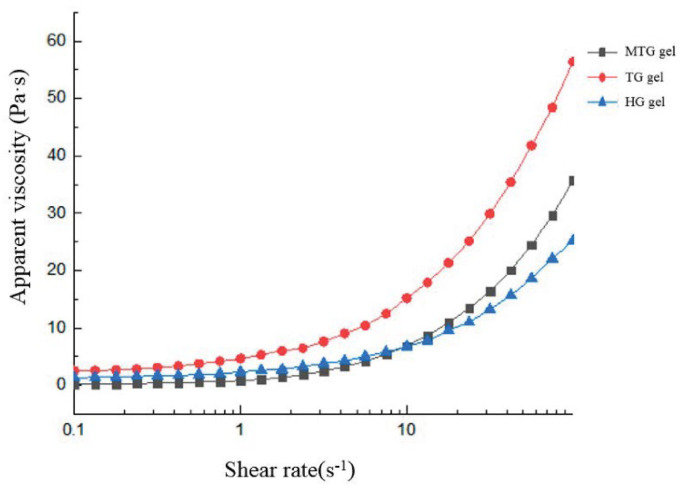
Flow behaviors (apparent viscosity versus shear rate) of HG gel, TG gel, and MTG gel at a protein concentration of 12% (*w*/*v*).

**Figure 7 foods-11-03595-f007:**
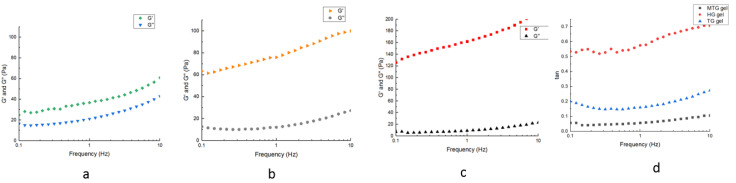
G’ and G’’ of different gels. Note: (**a**) for HG gels; (**b**) for TG gels; (**c**) for MTG gels; (**d**) for tan δ.

**Figure 8 foods-11-03595-f008:**
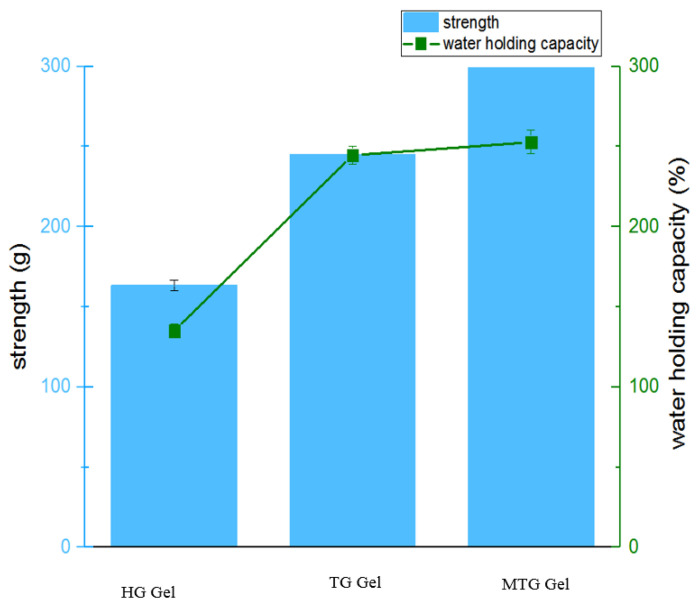
Gel strength and water-holding capacity of different induced gels.

**Table 1 foods-11-03595-t001:** Secondary structure content of different gels.

Samples	β-Fol (%)	β-Turn (%)	Random Coil (%)	α-Helices (%)
SPI	32.87 ± 0.17	13.22 ± 0.66	33.29 ± 0.35	30.44 ± 0.10
HG gel	36.86 ± 0.23	21.51 ± 0.77	23.36 ± 0.34	32.73 ± 1.25
TG gel	43.52 ± 0.74	21.80 ± 0.83	25.51 ± 1.22	28.65 ± 0.53
MTG gel	47.62 ± 0.93	23.88 ± 0.92	28.99 ± 1.02	29.18 ± 0.87

## Data Availability

The data is included in this article.

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
