# Peer review of "The Effect of Different Induction Methods on the Structure and Physicochemical Properties of Glycosylated Soybean Isolate Gels"

_foods, 2022, doi:10.3390/foods11223595_

Round 1
Reviewer 1 Report
Authors of paper entitled “Effect of different induction methods on the structure and physicochemical properties of glycosylated soybean isolate protein gels” designed a thoughtful experimental plan to analyze the influence of difference induction techniques on gel strength of SPI by endorsing the results by relevant analyzing techniques. However, this manuscript has been weakly conceptualized in presenting the results and discussing it deeply with relevant literatures to justify their observations (some has been pointed out as below). There are many flaws in results and discussions.
L 125-136: to make it easier for the reader, please add two formula for emulsifying activity and emulsion stability with description of its related elements in detail. I specifically refer to the Φ and c parameters and implication of constant value of 10000 along with protein concentration.
L 154-155: Figure 1 needs to be changed with high quality version. It also needs a thorough descriptive caption with molecular weight ladder.
Fig. 2: as indicated in this figure, SPI shows a higher absorbance (lower transmittance) compared to other induction treatments at 1540-1658. Does that implies on its ability to form a robust gel network?
L 176: …”showed an increase at 1658 cm-1 and 1540 cm-1,”… needs to be magnified in an embedded photo along with Fig.2 or be presented in a separate figure related to the secondary derivative spectra of different samples at amid I band.
L 181: Table 1- the superscript letters for Beta folding values are absent. It would be also very informative to add the secondary structure values for SPI to indicate the magnitude of secondary structure deviation after induction by different treatments.
L192-194: this statement is only valid for MTG gel and not TG.
195-204: the speculation by the authors regarding the increasing the randomness/disorder of system due to action of enzyme needs to be elaborated significantly by adding valid and updated references.
L 308: “Discussion” needs to be changed to “Conclusion”
L 249-277: Emulsifying and viscosity sections needs to be fundamentally modified and revised as the results have been conceptualized wrongly.
L 279-307: Dynamic rheological analysis has been poorly presented and discussed by only frequency sweep test while it would be more appropriate to present the temperature sweep results and relevant discussion.
Author Response
Thank you for comments concerning our manuscript entitled “The Effect of Different Induction Methods on the Structure and Physicochemical Properties of Glycosylated Soybean ” (ID: foods-1951821). Those comments are all valuable and very helpful for revising and improving our paper, as well as providing important guiding significance to our research. We have studied comments carefully and have made corrections, which we hope are met with approval. The main corrections in the paper and the responses to the Reviewers’ comments are as follows:

Reviewer 2 Report
The article deals with the ability of glycosylated SOY isolate (MGSI) to gel under three methods (thermal , enzymatic and enzymatic with cation catalysis. The authors evaluate the system from microscopic scale to macroscopic analysis through rheology.
The topic could be of interest as it is known that plant-based protein may suffer of functional properties.
The authors compare 3 methods of gelation of MGSI. First all the method used did not characterize gel structure (ex electrophorese).
They compare two processes of enzymatic gelation with or without cation as catalysis and they said that TGase-MGCL2 as a synergic effect when they compared the viscoelastic properties. But they did not use the same concentration of MGSI (5% or 10 % in the presence of cation see L81-L89) . The increase in viscoelastic properties could be due to this increase in concentration of protein.
L81-89 : why the conditions of gelation in the presence of CaCl2 are different ?
L 106 -113 There is no explanation of what is expected to be analysed. Even in the result part, it is not clear about the wavelength that are looking for to explain the conformation of the protein and the literature cited is not really meaningful.
L106 FTIR measurement It is not well explain on which product ? Gel or « solution of diluted gel »)
L 126 Emulsification : The Emulsification activity is not detailed and there is no reference to explain the calculation .
Moreover it is not really clear if it is a gel that have an emulsification properties or more precisely aggregate (particle gel) that have this properties ?
L137 Rheological measurement : The author describe the flow measurement but they did not give any information about the dynamic measurement (which stress or strain applied. Is it in the LVE domain …)
L143 : The author explain the water retention but it is not discuss in the main document may be in the supplementary data but it should be at least mentioned.
RESULT:
Fig 1 as for al l the figure , the legend should help to understand the figure by itself.
The Size of the the marker should be added , Are the four strands repetitions?
Fig2 : the colour used did not allow to well see difference between HG and TG Gel.
L176 the author describe an increase of the pic at 1658 and 1540 cm-1 for treated gels but it is a decrease and an increase in the 1000 cm-1 . In this part it is not well describe from which band they elucidate the a or b structure. Some information should be given.
L 206 The result of fluorescence could not be compare if the concentration of protein is not the same
Fig 4,5 and 6 should be merged
The emulsification analysis part :
« the addition of TG 252 enzymes may act to inhibit covalent cross-linking within or between MGSI molecules, 253 forming gels with larger relative molecular masses and thicker interfacial protein layers 254 that have difficulty interacting with small molecule oil droplets, and therefore become less 255 emulsifiable » This part is not really clear. TG enzyme could not inhibit covalent crosslink as it is its role?
Static rheology :
Fig 8 It is surprising that viscosity increase with the shear rate? It a shear thickening behaviour.
The author explain a dissociation of MGSI trough the treatment with TGase . It is surprising TGase promote covalent linkage? It is not coherent with the electrophorese where the author evidence higher MW for treated MGSI.
Dynamic rheology:
Usually the result are represented in log scale to better reflect the dependence to frequency and both G’ and G’’ should be drawn on the same figure to better evidence that G’ is much higher and G’’ and that we may consider a gel.
L 297-300 This part is not clear. The bound are not formed during the rheological measurement
The less the module is dependant to the frequency the more the gel structure is stable and strong .
Author Response
Thank you for your comments concerning our manuscript entitled “The Effect of Different Induction Methods on the Structure and Physicochemical Properties of Glycosylated Soybean ” (ID: foods-1951821). Those comments are all valuable and very helpful for revising and improving our paper, as well as providing important guiding significance to our research. We have studied comments carefully and have made corrections, which we hope are met with approval. The main corrections in the paper and the responses to the Reviewers’ comments are as follows:

Reviewer 3 Report
the work presented has good contribution to the field.
Author Response

(The authors gave the same response as above.)

Round 2
Reviewer 1 Report
The manuscript has been modified in a significant level based on the comments from reviewers to make it scientifically interesting for the readers. However, there are many speculation in the abstract that needs to be removed and replaced by the factual results from the study. For instance, "...Compared with thermal induction, the secondary structure of MGSI induced by TGase was more stable,..." How did you measured the stability?
"...unfolded most fully, forming multilayer gels double cross-linked by covalent bonds and ionic salt bridges with the tightest structure...." This is unclear and a pure assumptional statement. needs to be revised.
and there is no final statement in abstract on practical application of results from this study.
"... and the tertiary structure of MGSI unfolded and cross-linked more fully between proteins;..."How did you measure the extent of unfolding and cross-linking of tertiary structure?
Author Response
Dear Reviewer,
Thank you for your comments concerning our manuscript entitled “The Effect of
Different Induction Methods on the Structure and Physicochemical Properties of
Glycosylated Soybean ” (ID: foods-1951821). Those comments are all valuable and very helpful for revising and improving our paper, as well as providing important guiding significance to our research. We have studied comments carefully and have made corrections, which we hope are met with approval.

Reviewer 2 Report
Some improvement have been done on the article considering the first comment however some results and conclusions are still not comprehensive and may suffer of non-correct literature review (see comment on apparent viscosity as example). Some reference are not in accordance with the subject or the description( Ref 41, 42, ..)
It is not really understandable why the author did not measured the properties of the gels without dilution?
The author still did not explain why they use 10%MGSI instead of 5% when they use MgCl2-TG gelation process. The increase in the concentration comparing to the two other process may explain difference
The author used lyophilized gel powder : did they grind the lyophilized gel?
2.3.4 “The SPI and gel strength were adjusted to 40 mg/mL” what means strength?
2.3.7 “A total of 0.1 g of gel product was weighed with a protein concentration of 0.2%...” gel product : is it lyophilized gel powder ? The amount of protein should not be the same in all gel if 0.1g is taken for all gel as the MGCl2 -TG is done from 10 % protein?
2.3.8 “Referring to the method in the literature [17], the samples were prepared in distilled water to a concentration of 12% gel liquid” If 12% gel for the 3 gels , the gel with MGCL2 will be twice concentrated in protein?
“Measurement of storage modulus (G') and loss modulus (G''): A rotational viscome-ter was used to measure the rheological behavior of the emulsions”” Is it the rheology of emulsion or gel?
In the equation what is A10 and 0
“In the sweep mode, a constant strain of 0.5% was set and the loss modulus was measured The loss modulus and storage modulus of the gel were measured at frequencies from 0.1 to 10 Hz..”
2.3.9 How was the gel? (cylinder, cube…)
2.3.10 “The gel was cut into cylindrical shapes (20 mm diameter and 10 mm height) and the gel strength of the samples was determined using a TA-XT mass spectrometer. A P /36R probe was used with a trigger force of 0.0294 N, a drop height of 10 mm,” the gel is compressed on its all height?
Results :
3.1 : FTIR The HG and MGSI signal are not distinguished are they totally superposed in the range of the wavelength studied. The author should change the color it is not well define .
The authors did not comment the change in the region 1000- 1200 cm-1 were the difference ar drastic?
“The absorption peaks of MGSI protein gels were wider in the range of 3200 cm−1~3700 cm−1 compared to the three protein gels (HG, TG, , and MTG)” The HG and MGSI signals are superposed in this region? As in most of the region. The main differences are for TG and MGTG.
“Compared to SPI, the main structures in the three gel particles were still β-fold and α-helix, and the β-fold content showed an increasing trend. This indicates that on the one hand, induction has an effect on the secondary structure of the protein [23], and on the other hand, the introduction of the sugar chain causes the protein peptide chain to unfold, the internal groups to be exposed, and the spatial conformation of the protein to change” and Table 1 : There is no comparison with SPI but only with MGSI?
Figure 6 : The legend of the figure should be more explicite (dilute gel at 12%? ).
“The apparent viscosities of TG and MTG gels were higher than those of HG gels, probably due to the “expansion” of the volume of the gel particles as a result of the de-struction of the original structure and the formation of a new structure by external forces: the shear-thickening phenomenon. Previous studies have also shown that casein and whey proteins treated with enzymes or salt also exhibit an increase in apparent viscosity [41-42].”
The reference 41 and 42 did not deal with WPI or casein :
41. Zhang, L.; Cui, X.; Zhao, C.; Li, W; Chen, Z; Tang, L;Lai,F; Ai, L; Zhan, H. Effect of barley polysaccharides on the pasting and rheological properties of corn starch. J. Food Biotechnol. 2020, 39, 73–81.
42. Jiang, S.; Zhao, X. Transglutaminase-induced cross-linking and glucosamine conjugation in soybean protein isolates and its impacts on some functional properties of the products. Eur. Food Res. Technol. 2010, 231, 679–689.
The authors refer to the article of Jiang 2010 to say that the product are shear thickening. However in the article of Yang the author show a shear thinning behaviour. Are the author representing in the figure 6 the viscosity or the stress?
The ref 43 deal with emulsion and not gel? This is not a reference to be used for rheology explanation
The dynamic rheology is done on diluted sample?
Fig7 The author should measured the dependence to the frequency ( calculating the slope) to evidenced the lower dependence of the TG gel due to covalent bound as discuss by Tang.
The step increase in G ‘ and G’’ for HG at 10 hz could be due to to high frequency for such low gel
The author explain the increase in the properties in the presence of Mg2+ due to salt bridges? But is there any effect of Mg2+ on TGASe activity . This should be controlled or proved form the literature.
Author Response
Dear editors and reviewers,
On behalf of my co-authors, I would like to thank you very much for giving us an opportunity to revise our manuscript. We appreciate the positive and constructive comments and suggestions on our manuscript entitled “The Effect of Different Induction Methods on the Structure and Physicochemical Properties of Glycosylated Soybean ” (ID: foods-1951821). We carefully considered all issues mentioned in the editors’ and reviews’ comments; we have provided a point-by-point response , and have made the necessary revisions. We hope the revised manuscript is met with approval. We would like to express our sincere appreciation to you and reviewers for the insightful comments on our paper.
